# Psychological intervention for negative emotions aroused by COVID-19 pandemic in university students: A systematic review and meta-analysis

**Qing Wang[1], Senbo An[2], Zebin Shao** [1] *

**1** School of Education Science, Nanjing Normal University, Nanjing, China, **2** Department of Orthopaedics, Shandong Provincial Hospital Affiliated to Shandong First Medical University, Jinan, China

* shaozebin2022@163.com

**Data Availability Statement:** Data sharing is not applicable to this article as no datasets were generated or analyzed during the current study.

## Abstract

The COVID-19 pandemic has been suggested to cause psychological problems such as anxiety, depression, panic, and insomnia. This systematic review and meta-analysis aims to assess the efficacy of psychological interventions (including CBT, DBT, and mindfulness based interventions) in reducing distressing feelings in college students during the COVID-19 pandemic. Randomized controlled trials (RCTs) on psychological interventions for reducing negative emotions among college students during the COVID-19 epidemic were searched in databases Web of Science, PubMed, Cochrane Library, Embase, Scopus, PsychInfo, CNKI, WanFang, and VIP. We utilized Cochrane risk of bias assessment technique to assess the quality of included RCTs. The data were analyzed using RevMan 5.4. Eight RCTs were finally included involving 1,496 participants. According to the meta-analysis results, psychotherapies could significantly alleviate anxiety, depression, and stress symptoms among university students, [MD = -0.98, 95%CI (-1.53, -0.43), p<0.001] [SMD = -0.47, 95%CI (-0.77, -0.16), p = 0.003] [MD = -1.53, 95%CI (-2.23, -0.83), p <0.001]. The difference in attenuation of somatization symptoms between the two groups was not statistically significant [SMD = -0.42, 95%CI (-1.41, 0.56), p = 0.40]. Psychotherapy might effectively alleviate anxiety, depression, and stress in university students caused by the COVID-19 pandemic. It could be considered the preferred strategy for reducing negative emotions and improving the quality of life of university students.

## 1 Introduction

Corona Virus Disease 2019 (COVID-19), a potentially deadly disease caused by SARS-CoV2, presents a significant problem facing global public health [1,2]. It not only poses a great threat to public physical health, but also causes negative emotions such as anxiety, depression, panic, and insomnia [3,4], leading to emotional breakdown and a sense of being abandoned, desperation, incapability, and exhaustion, even resulting in suicide [5]. It is prominent that university

**Funding:** This work was supported by the National Natural Science Foundation of China (Grant No.82102600), The funders had no role in study design, data collection and analysis, decision to publish, or preparation of the manuscript.

**Competing interests:** The authors have declared that no competing interests exist.

students would be more susceptible to psychological problems during the pandemic [6]. In addition to the pandemic, they would undergo a long and relatively closed special holiday and would cope with physical and mental discomforts when they returned to school, leading to varying degrees of worries about their emotional experience, academic performance, and life. They would also be anxious or concerned because they were afraid of being infected [7–9]. Though anxiety for wellbeing is a widely observed phenomenon, there is evidence proving that such anxiety could be elevated and more universalized during the pandemic [10,11]. Studies have shown poorer overall mental health status in university students during the pandemic. The overall incidence of anxiety and depression in university students is 26.60% and 21.16%, respectively [9]. Overwhelming negative feelings brought on by the COVID-19 pandemic may decrease their immunity [12], rendering them more vulnerable to infection. Therefore, addressing these negative emotions has become an urgent problem.

Psychological treatment primarily attempts to promote treatment motives, improve patients' self-confidence and self-efficacy, treat mental and behavioral problems, and support patients in learning various psychological skills to help them live a healthy lifestyle [13,14]. Psychotherapies that are included in this study contain cognitive-behavioral therapy (CBT), mindfulness-based cognitive therapy (MBCT), and dialectical behavior therapy (DBT). The idea of "mindfulness" is derived from Buddhist meditation, and is developed from meditation, dhyana, and consciousness in Buddhism. J. Kabat-Zinn et al [15]. defined it as a method of mental training. In this kind of mental training, it is emphasized to perceive consciously, focus on "here and now," and assess no ideas of the moment. CBT is a short course of psychotherapy that uses thinking and behavior changes to change perceptions and eliminate negative emotions and behaviors and focuses on thought and psychological education in university students to make them aware of the emotional experiences they are having to make early interventions to avoid these emotions from recurring [16]. DBT is a highly structured therapy consisting of individual psychotherapy, group skill training, telephone coaching, and a therapist group, each unit having unique therapeutic goals and the strategies necessary to accomplish those goals and intended to treat people who have extreme behavior disorders, and its rationale is to eliminate extreme behaviors by learning the "majjhima-patipada" to achieve "balance." [17,18] It is of great importance to take necessary interventions for these negative emotions induced by the pandemic. In recent years, many systematic reviews have been published on the effectiveness of psychological intervention on psychological problems caused by COVID-19 [19,20]. However, previous studies have not focused on the special group of college students and the effectiveness of psychological interventions is undergoing debate. The objective of the meta-analysis to investigate the efficacy of the psychological intervention in alleviating negative emotions sparked by the COVID-19 pandemic in university students and to provide reference for the treatment of psychological issues in university students.

## 2 Methods and materials

### 2.1 Exclusion and inclusion criteria

**2.1.1 Types of included literature.**   Randomized controlled trials (RCTs).

**2.1.2 Inclusion criteria.**   University students were afflicted by negative emotions during the COVID-19 pandemic. Psychotherapies (including mindfulness, CBT, and DBT) were applied as interventions in the experimental group, while wait-list, conventional nursing, etc., were set as control. Outcome measures included anxiety, depression, stress, somatization symptoms, etc.

Conference summary, non-human study, literature review, repeated publication, and study with incomplete data or data available were all excluded.

## 2.2 Literature screening

The literature search was performed based on medical subject headings, which mainly included: COVID-19, college students, university students, mindfulness, cognitive behavior therapy, and dialectical behavior therapy. RCTs were searched in databases Embase, Cochrane, PubMed, Web of Science, Scopus, PsychInfo, China National Knowledge Infrastructure (CNKI), VIP, and WanFang, from inception to April 1st, 2022, See the supplementary materials for specific search strategies.

## 2.3 Information extraction

The literature was screened independently by two reviewers, and then, the two reviewers would cross-check. Any disagreement was addressed by the third reviewer. After duplicate-checking, papers with irrelevant content were removed after reading titles and abstracts. Finally, eligible studies were identified by screening the full text. Key information extracted included: the publication date, the study's design, the first author's name, the country, the average participant age, the sample size, the intervention and control conditions, the length of the follow-up period, and outcome indicators.

## 2.4 Assessment of risk of bias

Two reviewers independently evaluated the quality of eligible studies via the Risk of Bias Assessment Tool in *Cochrane Handbook for Systematic Reviews of Interventions 5.1.0*. The following parameters were taken into account for the quality assessment: generation of a randomized sequence (selection bias), concealment of allocations (selection bias), blinding of participants and personnel (performance bias), blinding of outcome reviewers (detection bias), selective outcome reporting (reporting bias), incomplete outcome data (attrition bias), and other sources of bias. A study was rated to have a "low-risk" if it met all of the above criteria, "unclear-risk" if it only met some of them, and "high-risk" if it did not satisfy any of them.

## 2.5 Statistical analysis

Software RevMan 5.4 was utilized to conduct the meta-analysis. The heterogeneity test was performed via the Chi-square test. The fixed-effects model was employed if $p \geq 0.1$ with an I2 <50% which indicated the existence of statistical homogeneity among included studies; otherwise, the random-effects model would be utilized. And then a subgroup analysis was performed according to the types of psychological interventions to determine heterogeneity sources. For continuous data with the same measurement unit, mean difference (MD) was adopted as a pooled statistic, whereas Effect Size measurement (such as Cohen's D) standardized mean difference (SMD) was applied for those with different units, and the 95% confidence interval (95%CI) for each effect was provided. The funnel plot was employed to examine publication bias. A symmetrical one indicated that no significant publication bias existed.

# 3 Results

## 3.1 Study screening results

We identified 339 articles after an initial search in databases, and 259 of them were determined to be eligible after checking duplicate publications. We excluded 239 articles by screening titles and abstracts. And 12 were excluded during screening the full texts. Finally, 8 RCTs were considered to be eligible for this meta-analysis. **Fig 1** illustrates the literature screening process.

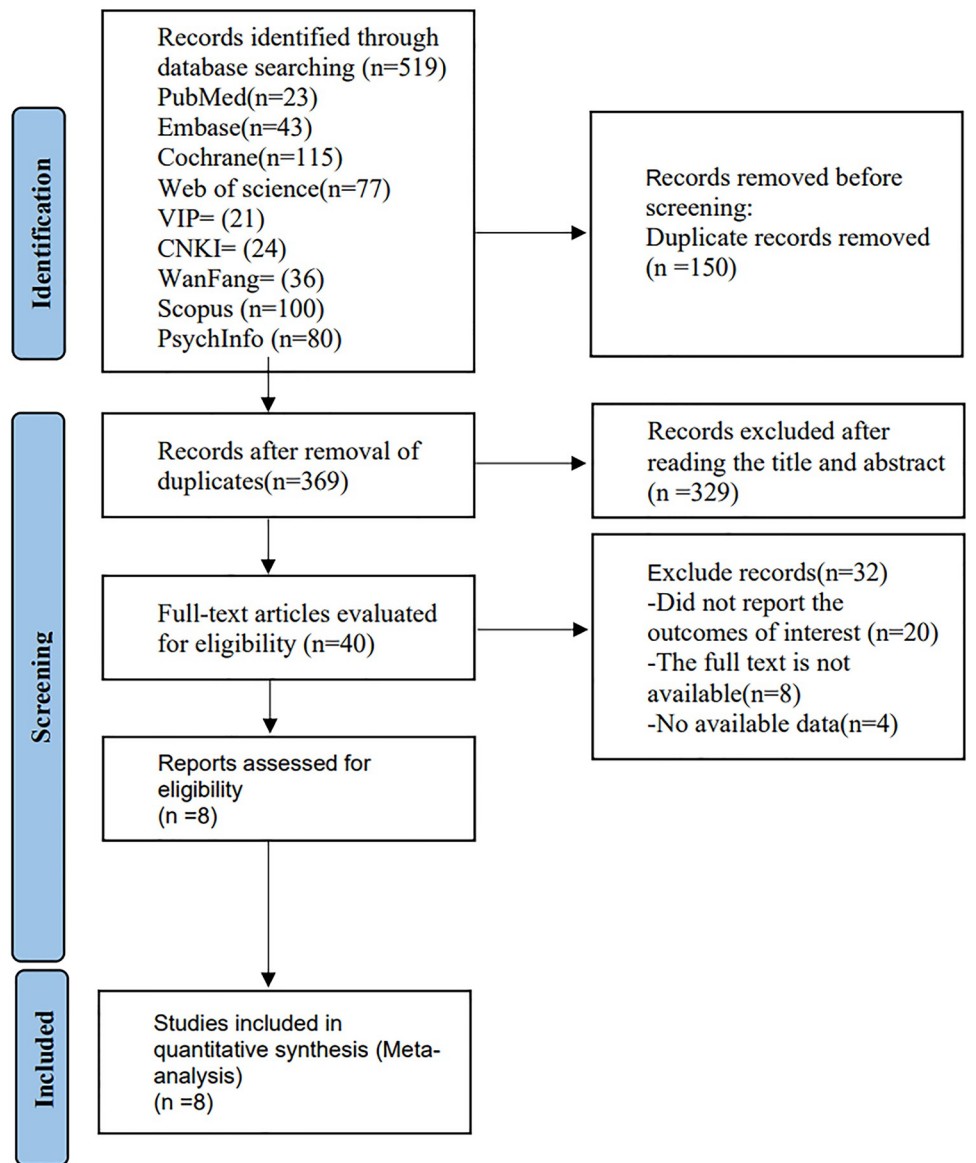

**Fig 1. Flow diagram of study selection.**

## 3.2 Study characteristics

Eight eligible RCTs [21–28] gave us a sample size of 1,496 subjects (755 in the experimental group, and 741 in the control group). Two [21,22] of the investigations were carried out in China, four [24–27] in the United States, one [28] in Iran, and one [23] in Canada. As for intervention, 5 [21,23–26] studies applied mindfulness-related psychotherapies, one [22] used DBT, 1 [27] used CBT+MBSR, and another 1 [28] applied CBT (as shown in Table 1).

Two assessors independently evaluate the quality of included RCTs via the Risk of Bias Assessment Tool from *Cochrane Handbook for Systematic Reviews of Interventions 5.1.0.* Two articles did not explain the specific randomized method, so they were rated as unclear. Moreover, these two studies did not mention the blind method used at all, so they were rated as high risk. The specific risk of bias assessment is shown in Figs 2 and 3.

**Table 1. Presents the characteristics of included RCTs.3.3 Risk of bias results.**

| First author, year | Country | Study design | Age(year) M(SD) | | Number of participants (Male) | | interventions | | Follow-up (Week) | Outcomes measures |
|---|---|---|---|---|---|---|---|---|---|---|
| | | | EG | CG | EG | CG | EG | CG | | |
| Li Yang 2020 [21] | China | RCT | 18.72 (0.66) | 18.49 (0.81) | 53(24) | 51(25) | MBSR | Blank control | 1 | F1; F2; F3; F4; F5; |
| LZ Liang 2021 [22] | China | RCT | 20.73 (1.87) | 20.62 (1.79) | 26(10) | 26(10) | DBT | Blank control | 4 | F3; F4; F5; F6; |
| C. El Morr 2021 [23] | Canada | RCT | 22.8(6.4) | 22.3(5.9) | 79(22) | 80(10) | MVC | Blank control | 8 | F4; F5; F7; |
| O. Simonsson 2021 [24] | USA | RCT | 18–55 | 18–55 | 88(26) | 89(30) | MBSR | Blank control | 8 | F3; F4 |
| P. Ritvo 2021 [25] | USA | RCT | 22.02 (5.52) | 24.18 (9.95) | 76(18) | 78(17) | MVC | Blank control | 8 | F3; F4; F5; F7; |
| SF Sun 2022 [26] | USA | RCT | 22.1(2.67) | 22.1(2.67) | 57 | 57 | MBSR | Social Support | 4 | F3; F4 |
| G. N. Rackoff 2022 [27] | USA | RCT | 20.24 (4.13) | 20.65 (4.88) | 301 (65) | 284 (52) | CBT +MBSR | usual care | 12 | F3; F4; F5 |
| R. Shabahang 2021 [28] | Iran | RCT | 24.7(5.4) | 24.7(5.4) | 76 | 76 | CBT | Blank control | 3 | F4; F6 |

RCT: Randomized controlled trial; EG: Experimental group; CG: Control group; MBSR: Mindfulness-based stress reduction; DBT: Dialectical behavior therapy; MVC: Mindfulness Virtual Community; CBT: Cognitive-behavioral therapy; F1: Pittsburgh Sleep Quality Index (PQSI); F2: Mood state Scale (POMS); F3: Depression; F4: Anxiety; F5: Stress; F6: Somatization symptoms; F7: Five Facets Mindfulness Questionnaire Short Form.

## 3.3 Meta-analysis outcomes

**3.3.1 Depression.** Five studies [21,22,24,26,27] reported depression, involving 1,156 subjects (585 in the experimental group and 571 in the control group). We used the fixed-effects model due to no significant heterogeneity across the studies ($p = 0.84$, $I^2 = 0\%$). Outcomes of the meta-analysis illustrated that participants in the psychological intervention group had lower depression symptom scores than those in the control group, and the difference was statistically significant [MD = -0.98, 95%CI (-1.53, -0.43); $p<0.001$], as shown in **Fig 4**.

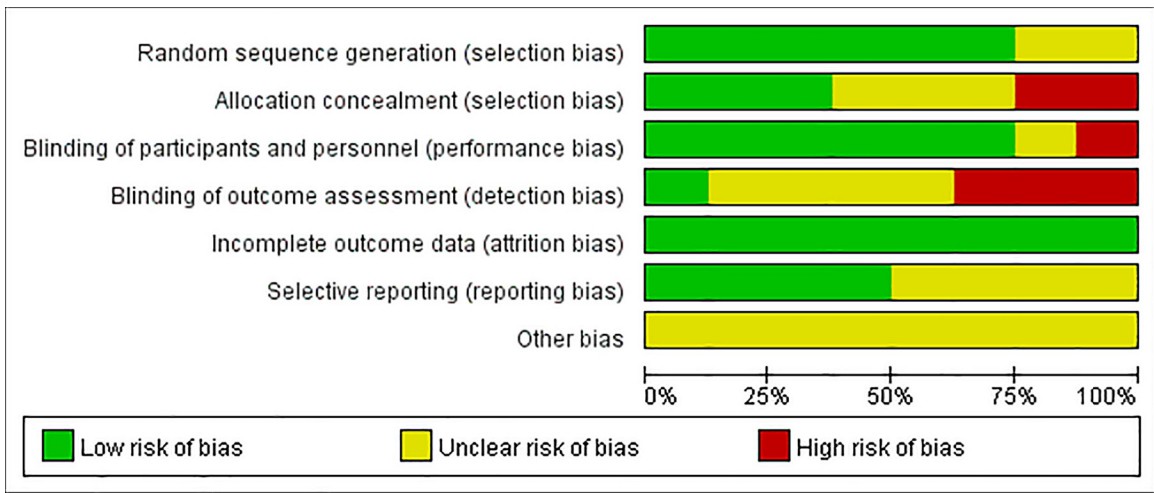

**Fig 2. Risk of bias graph.**

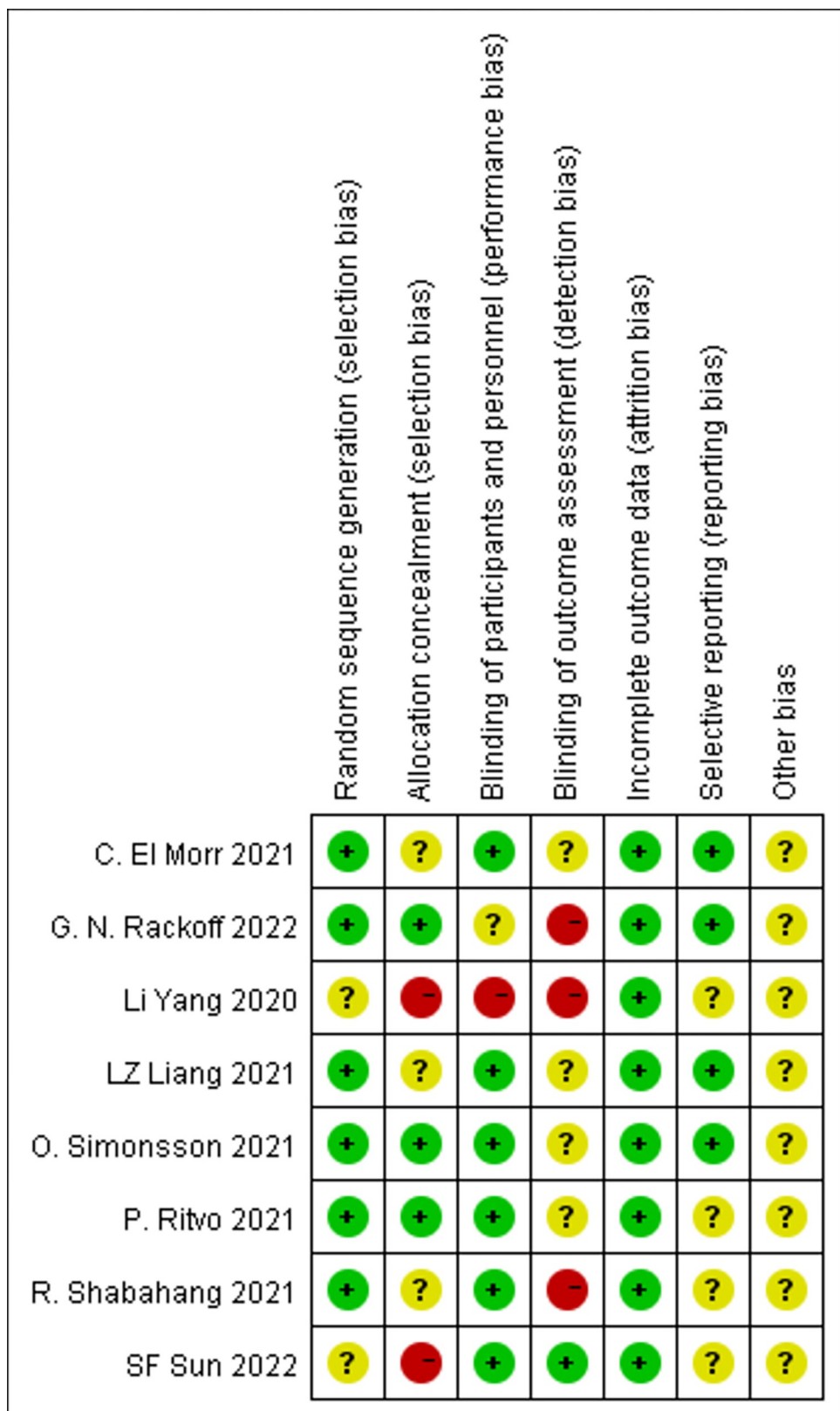

**Fig 3. Risk of bias summary.**

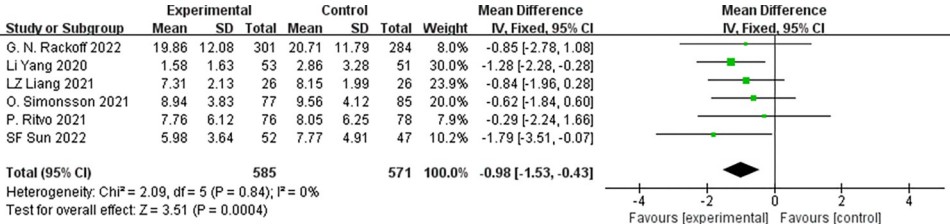

**Fig 4. Forest plot for depression score.**

**3.3.2 Anxiety.** Eight studies [21–28] reported anxiety, involving 1,454 subjects (728 in the experimental group and 726 in the control). We utilized the random-effects model due to the existence of significant heterogeneity across the studies ($p<0.001$, $I^2 = 84\%$). Outcomes of the meta-analysis illustrated that subjects in the psychological intervention group had lower anxiety symptom scores than those in the control group. There was a statistically significant difference [SMD = -0.47, 95%CI (-0.77, -0.16); $p = 0.003$], as shown **Fig 5**.

**3.3.3 Stress-related symptoms.** Five studies [21–23,25,27] reported stress-related symptoms, involving 1,042 participants (523 in the experimental group and 519 in the control). We adopted the fixed-effects model due to no significant statistical heterogeneity existing across the studies ($p = 0.93$, $I^2 = 0\%$). According to the outcomes of the meta-analysis, subjects in the psychological intervention group had lower stress symptom scores than those in the control group, and the difference was statistically significant [MD = -1.53, 95%CI (-2.23, -0.83), $p<0.001$], as shown in **Fig 6**.

**3.3.4 Somatization symptoms.** Two studies [22,28] observed somatization symptoms, involving 202 subjects (101 in the experimental group and 101 in the control). We employed the random-effects model due to the existence of significant heterogeneity across the studies ($p = 0.002$, $I^2 = 90\%$). In terms of reducing somatization symptoms, the meta-analysis outcomes illustrated no statistically significant difference [SMD = -0.42, 95%CI (-1.41,0.56), $p = 0.40$] (**Fig 7**).

**3.3.5 Subgroup analysis.** Subgroup analysis was performed for the anxiety score that showed significant heterogeneity, and subgroups were set based on different psychological interventions. There was significant heterogeneity amongst the trials ($p<0.001$, $I^2 = 86\%$), so we employed the random-effects model for analysis. The results illustrated that mindfulness intervention was more effective in decreasing anxiety symptom scores, with statistical significance [SMD = -0.41, 95%CI (-0.65, -0.16), $p = 0.001$]. With regard to CBT, the difference in the reduction of anxiety symptom scores between the CBT group and the control group was statistically significant [SMD = -0.51,95%CI (-1.59, 0.58), $p = 0.38$]; however, as for DBT intervention, the difference in the reduction of anxiety symptom scores between two groups was statistically significant [SMD = -0.77, 95%CI (-1.33, -0.20), $p = 0.008$] (**Fig 8**).

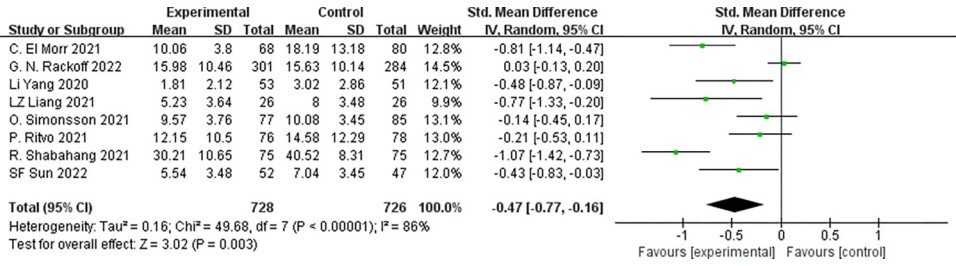

**Fig 5. Forest plot for anxiety score.**

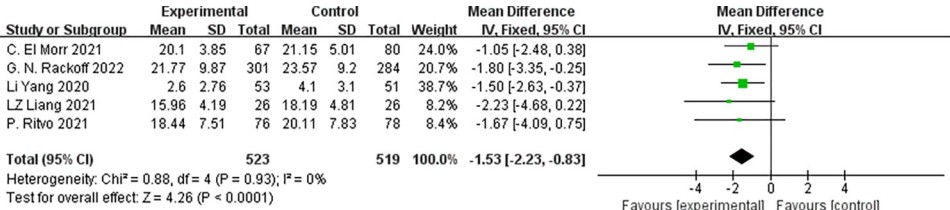

**Fig 6. Forest plot for stress-related symptom score.**

**3.3.6 Sensitivity analysis and examination of publication bias.** We conducted the egger test for each anxiety, depression, stress and other indicators, and found that anxiety p = 0.046, depression p = 0.801, stress p = 0.355, P>0.05, indicating that there was no publication bias, and P<0.05, indicating that there was publication bias. (**Fig 9A, 9B and 9C**). The forest plot illustrated that outcome measure of anxiety symptom scores had significant heterogeneity. Meanwhile, the removal of studies one by one did not reverse the results, suggesting that the bias caused by a single study was minimal, and the meta-analysis results were robust.

# 4 Discussion

The suddenness and seriousness of the pandemic have taken the general people by surprise. The local administration has implemented many restrictive anti-epidemic policies (restrictions on going out, relatives visiting, gathering, etc.) [29], though these measures bring disturbance to people's normal life. The increasing number of diagnosed cases and deaths, infected loved ones, and long-lasting social isolation make the public subjected to various physical and psychological problems. The occurrence of anxiety and depression is often accompanied by a certain degree of inflammatory reaction, which is the change of patients' immune function. Clinical studies have found that compared with normal people, patients with depression have proinflammatory cytokines such as tumor necrosis factor-α(TNF-α), Interleukin-1β(IL-1β), IL-6 and interferon While anti-inflammatory cytokines such as IL-10, IL-4, IL-8 and transforming growth factor. After the use of antidepressants, its expression was up-regulated. After injection of lipopolysaccharide (LPS), normal people will soon show symptoms of depression or anxiety, and its severity is related to the increased expression of proinflammatory cytokines in blood. The mechanism may be that proinflammatory cytokines enter the brain through the blood brain barrier and interfere with the transmission function of neurotransmitters, thus affecting the mental activity of the body, leading to depression.

This study indicated that psychological intervention might be beneficial in relieving COVID-19 pandemic-induced anxiety and depression in university students, which echoes the findings of Liu et al. [29]. Their study involved 51 patients who were divided into the control and experimental group. All patients had completed a self-assessment questionnaire before entering an isolated ward. Patients in the experimental group listened to an audio of mindfulness before the noon break and during sleep. Patients in the experimental group experienced

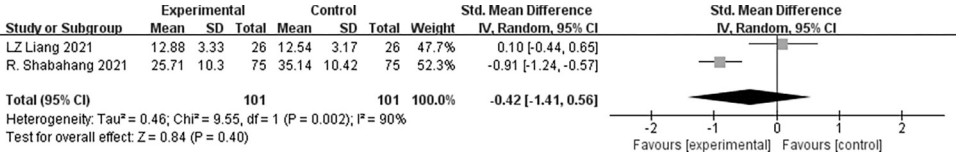

**Fig 7. Forest plot for somatization symptom.**

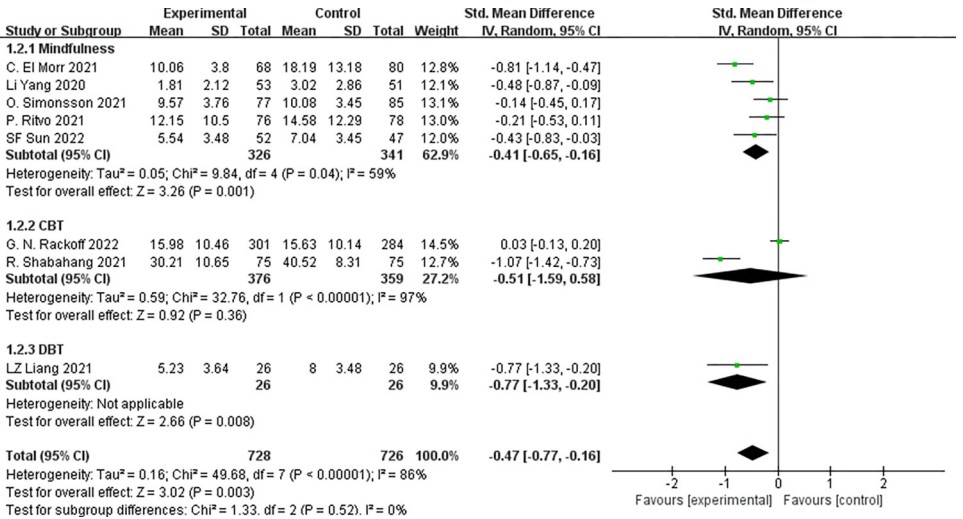

**Fig 8. Subgroup analysis for anxiety.**

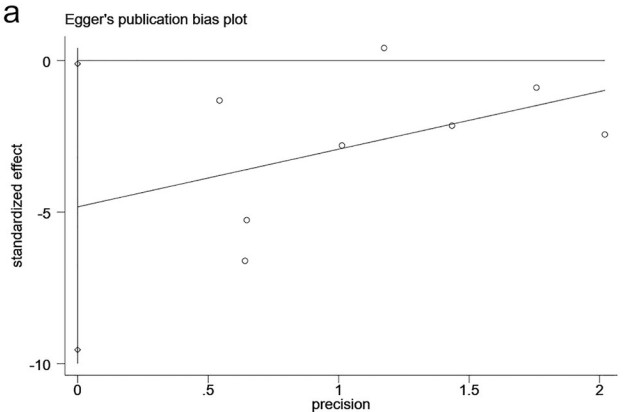

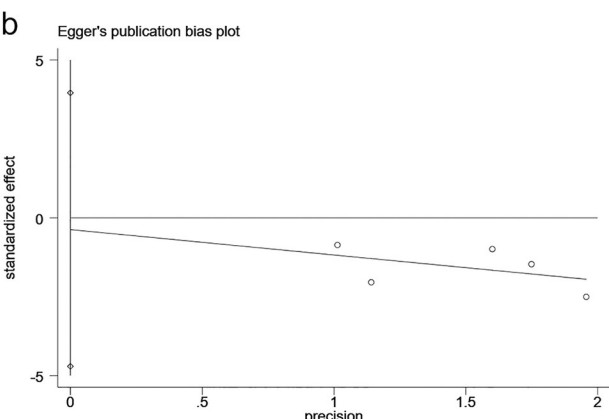

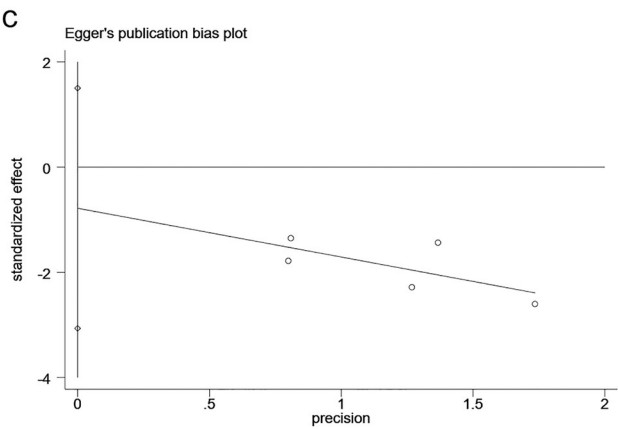

**Fig 9.** A: The egger test of anxiety; B: The egger test of depression; C: The egger test of stress-related symptoms.

much less anxiety than those in the control group. By diverting their attention from the negative feelings brought on by the pandemic, mindfulness assisted patients in refocusing on their breathing. On the other hand, holding an attitude of mindfulness-accepting and non-judgment could also help prevent those negative emotions [30]. Several studies have confirmed that the activity of the θ wave significantly increases in patients who received mindfulness training for a long time, indicating increased alertness of the brain and improved attention and cognition [31], as well as enhanced ability of emotion regulation and better life quality. In contrast to the findings of Marissa et al. [32], our meta-analysis outcomes illustrated that the difference in somatic symptoms between two groups was not statistically significant. This might be associated with the number of studies included. In addition to reducing the specific clinical symptoms of negative emotion, mindfulness meditation training could also regulate the activity of related brain regions, mainly involving the dorsomedial prefrontal cortex (DMPFC) and the dorsolateral prefrontal cortex (DLPFC) [33].

This study has demonstrated that psychological intervention is effective in alleviating negative emotions such as anxiety, depression, and stress. However, there are still some limitations. Firstly, the scales used to assess depression and anxiety varied between studies, resulting in significant heterogeneity. Secondly, the number of included studies was limited in that only Chinese and English databases were searched. Lastly, different psychological interventions are included in this study contributing to heterogeneity among included studies.

## 5 Conclusion

Psychotherapy might effectively alleviate depression, anxiety, and stress in university students caused by the COVID-19 pandemic. It could be considered the preferred strategy for reducing negative emotions and improving the quality of life of university students. However, more multi-center RCTs with high quality and large sample sizes are needed due to the small number of studies include in this meta-analysis.

## Supporting information

**S1 Checklist. PRISMA 2020 checklist.**
(DOCX)

**S1 Appendix. Search strategies.**
(CSV)

## Author Contributions

**Conceptualization:** Qing Wang.

**Data curation:** Qing Wang.

**Formal analysis:** Qing Wang.

**Funding acquisition:** Qing Wang.

**Investigation:** Qing Wang, Senbo An.

**Methodology:** Qing Wang, Senbo An.

**Software:** Qing Wang.

**Supervision:** Zebin Shao.

**Validation:** Senbo An, Zebin Shao.

**Visualization:** Qing Wang.

**Writing – original draft:** Qing Wang.

**Writing – review & editing:** Qing Wang, Senbo An, Zebin Shao.

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
