## [Decision Letter · Decision Letter 0]

2 Nov 2022

PONE-D-22-25934Psychological Intervention for Negative Emotions Aroused by COVID-19 Pandemic in University Students: A Systematic Review and Meta-AnalysisPLOS ONE

Dear Dr. Zebin Shao,

Thank you for submitting your manuscript to PLOS ONE. After careful consideration, we feel that it has merit but does not fully meet PLOS ONE’s publication criteria as it currently stands. Therefore, we invite you to submit a revised version of the manuscript that addresses the points raised during the review process.

Please revise your paper. Please follow the reviewers' suggestions below.

We look forward to receiving your revised manuscript.

Kind regards,

Gabriella Vizin, PhD

Academic Editor

PLOS ONE

Journal Requirements:

"This work was supported by the National Natural Science Foundation of China (Grant No.82102600)."

"This work was supported by the National Natural Science Foundation of China (Grant No.82102600)"

"This work was supported by the National Natural Science Foundation of China (Grant No.82102600)."

4. PLOS requires an ORCID iD for the corresponding author in Editorial Manager on papers submitted after December 6th, 2016. Please ensure that you have an ORCID iD and that it is validated in Editorial Manager. To do this, go to ‘Update my Information’ (in the upper left-hand corner of the main menu), and click on the Fetch/Validate link next to the ORCID field. This will take you to the ORCID site and allow you to create a new iD or authenticate a pre-existing iD in Editorial Manager. Please see the following video for instructions on linking an ORCID iD to your Editorial Manager account: https://www.youtube.com/watch?v=_xcclfuvtxQ.

Reviewers' comments:

Reviewer's Responses to Questions

**Comments to the Author**

1. Is the manuscript technically sound, and do the data support the conclusions?

Reviewer #1: Yes

Reviewer #2: Yes

2. Has the statistical analysis been performed appropriately and rigorously? 

Reviewer #1: Yes

Reviewer #2: Yes

3. Have the authors made all data underlying the findings in their manuscript fully available?

Reviewer #1: Yes

Reviewer #2: Yes

4. Is the manuscript presented in an intelligible fashion and written in standard English?

Reviewer #1: Yes

Reviewer #2: Yes

5. Review Comments to the Author

Reviewer #1: Review about the manuscript entitled as ‘Psychological intervention for negative emotions aroused by COVID-19 pandemic in university students: a systematic review and meta-analysis

Dear authors, thank you for allowing me to review your work.

Summary and Overall evaluation

The study is addressed to the systematic and metanalytic review of previous studies reporting results about psychological interventions conducted in randomized controlled trials for negative emotions induced by COVID-10 pandemic in university students. The authors considered 259 articles for review based on extensive database search, and selected finally 8 studies for the full review. The study used appropriate eligibility criteria for study selection and used adequate metanalytic techniques to analyse the effectiveness of the interventions. The analyses performed suggested a positive influence of psychological interventions on emotional complaints arisen during the COVID crisis. More specifically, the authors concluded that the studies reviewed provided clear evidence for significant effectiveness of psychotherapies in alleviating the anxiety, depression, and stress, but not the somatization symptoms. The study was overall well written; I do not have sever concerns. Please see my detailed comments and questions below.

Comments & Questions

[1] In recent years, many systematic reviews have been published on the effectiveness of psychological interventions for psychological problems caused by the COVID epidemic. How different and specific is the present review compared to the previous ones with similar scope? I suggest that the authors extend the introduction with a comparison of their study with similar reviews published earlier.

[2] Based on the supplementary information (i.e. PRISMA checklist) and the flow diagram of Figure 1, the authors followed the PRISMA guidelines to select the relevant studies. This, however, should also be stated explicitly in the Method section by adding reference(s) relating to PRISMA; e.g. the following one:

Grimshaw et al (2021). The PRISMA 2020 statement: An updated guideline for reporting systematic reviews. Systematic Reviews, 10(1), 89. https://doi.org/10.1186/s13643-021-01626-4

[3] The authors report that many databases were searched for the relevant studies. The general or database-specific search strings (syntax), however, are not reported. The authors may consider adding the search strings either to the main text or the Supplementary material.

[4] In the Results section, I found the summary of findings regarding the participants tested in the different studies a bit short. I suggest a more detailed elaboration of the samples tested in the previous studies. Were there under-, or post-graduate students tested? Do the studies reviewed report any information about the students’ education type? I feel this question relevant, because, for example, medical students may have been even more affected by the pandemic than students enrolling other type of courses.

[5] According to the inclusion criteria only those studies were selected where the students tested were afflicted by negative emotions during the COVID-19 pandemic. Is there any information that the negative emotions encountered were specifically related to COVID-19? Were, for example, any COVID-specific anxiety or depression assessments made in the studies reviewed, or the emotions were assessed by general methods (e.g. general depression, anxiety inventories) only?

[6] The COVID-19 pandemic is typically characterized with consecutive waves. If this information is available in the 8 studies reviewed, it may also be informative to indicate under which COVID wave(s) the psychological interventions were performed.

[7] The authors used an appropriate method – the Cochrane Handbook for Systematic Reviews of Interventions – to assess the potential risk of biases in the studies reviewed. Although the results of the quality assessment are clearly presented in Figure 2-3, the authors should also summarize their findings for quality and bias assessment in text (e.g. in the same section where the findings for publication bias are reported).

[8] Finally, regarding the Discussion section, I have one question only. The authors explain the finding of the non-significant result for the somatic symptoms with that of “this might be associated with the number of studies included”. I found this statement unclear. Why would the number of studies included affect the intervention-outcome for somatic symptoms, or more specifically, why would it affect somatic symptoms more than other symptoms (depression and anxiety)?

Reviewer #2: Overall:

Looking into the emotional effects of Covid-19. This is something not often spoken about in society and something which holds merit and importance. Particularly the idea of abandonment, desperation, incapability and exhaustion.

Looking into the university age demographic and how they are more susceptible to the emotional effects of Covid-19 due to the different conditions when at school and away from school.

The theory proposed here is interesting and has the potential to show clear signs that psychotherapy has a positive impact on helping deal with the emotional strain of the Covid-19 pandemic. However, I think that parts of this theory are generic and rather broad. The theory suggests that psychotherapy could help with symptoms such as depression and anxiety brought on by the pandemic. Whilst this does hold some merit, I feel like this is true of depression and anxiety which is caused by any number of reasons besides the pandemic. It has been widely accepted that psychotherapy has a positive impact on anxiety and depression as a whole and I would have liked a little more data on the difference between the anxiety and depression felt as a result of the pandemic against the anxiety and depression felt in everyday life. Is the end result of anxious and depressed feelings still the same as before the pandemic or are we talking about a new and differing feeling as a result of Covid-19?

The link between anxiety and depression with the increase of infection. This was mentioned but not in depth and I think a little more detail regarding this would have helped give the paper more meaning and depth.

Meta-analysis/ results/method:

The analysis itself was well undertaken. I'd recommend that the offers incorporate the following into the paper to strengthen the work itself but otherwise I find it well written:

Overall, I would have suggested that the authors use another Effect Size measurement (such as Cohen's D) or Hedge's G for the measurement. Since it is not used here, i would employ authors to explain why it is not used.

-I'd recommend authors employee the Begg and Egger tests to further elaborate on the study findings. The Begg and Egger test usually is done after an asymmetrical funnel plot is undertaken. Also, if authors use such you can find how many studies are needed to negate the present effect.

-More within study differences would have been nice to see and I recommend authors add those in to the analysis. For example, authors could have seen the effect of gender, sex, medication status, education level, differences between countries, comorbidites. These could have strengthened the paper.

-I'd also recommend that the authors add in Scopus and PsychInfo as a way for them to add in more studies. As well as cross-checking between articles for more studies

6. PLOS authors have the option to publish the peer review history of their article (what does this mean?). If published, this will include your full peer review and any attached files.

Reviewer #1: No

Reviewer #2: **Yes: **Mara Czegel

---

## [Author Response · Author response to Decision Letter 0]

10 Nov 2022

List of Responses

Dear Editors and Reviewers:

Thank you for your letter and for the reviewers’ comments concerning our manuscript entitled “Psychological Intervention for Negative Emotions Aroused by COVID-19 Pandemic in University Students: A Systematic Review and Meta-Analysis” (ID: PONE-D-22-25934). Those comments are all valuable and very helpful for revising and improving our paper, as well as the important guiding significance to our researches. We have studied comments carefully and have made correction which we hope meet with approval. Revised portion are marked in red in the paper. The main corrections in the paper and the responds to the reviewer’s comments are as flowing:

Responds to the reviewer’s comments:

Review Comments to the Author

Reviewer #1

Review about the manuscript entitled as ‘Psychological intervention for negative emotions aroused by COVID-19 pandemic in university students: a systematic review and meta-analysis

Dear authors, thank you for allowing me to review your work.

Summary and Overall evaluation

The study is addressed to the systematic and metanalytic review of previous studies reporting results about psychological interventions conducted in randomized controlled trials for negative emotions induced by COVID-10 pandemic in university students. The authors considered 259 articles for review based on extensive database search, and selected finally 8 studies for the full review. The study used appropriate eligibility criteria for study selection and used adequate metanalytic techniques to analyse the effectiveness of the interventions. The analyses performed suggested a positive influence of psychological interventions on emotional complaints arisen during the COVID crisis. More specifically, the authors concluded that the studies reviewed provided clear evidence for significant effectiveness of psychotherapies in alleviating the anxiety, depression, and stress, but not the somatization symptoms. The study was overall well written; I do not have sever concerns. Please see my detailed comments and questions below. 

[1] Comment: In recent years, many systematic reviews have been published on the effectiveness of psychological interventions for psychological problems caused by the COVID epidemic. How different and specific is the present review compared to the previous ones with similar scope? I suggest that the authors extend the introduction with a comparison of their study with similar reviews published earlier.

Response: We are very grateful for the reviewer's comments. As the reviewer said, there have been many systematic reviews on the effectiveness of psychological intervention on psychological problems caused by COVID-19 in recent years, but previous studies did not pay attention to the special group of college students, and no similar meta-reports have been published.

[2] Comment: Based on the supplementary information (i.e. PRISMA checklist) and the flow diagram of Figure 1, the authors followed the PRISMA guidelines to select the relevant studies. This, however, should also be stated explicitly in the Method section by adding reference(s) relating to PRISMA; e.g. the following one:

Grimshaw et al (2021). The PRISMA 2020 statement: An updated guideline for reporting systematic reviews. Systematic Reviews, 10(1), 89. https://doi.org/10.1186/s13643-021-01626-4

Response: We strongly agree with the reviewer's opinion. We have rewritten prisma and attached it to the attachment.

[3] Comment: The authors report that many databases were searched for the relevant studies. The general or database-specific search strings (syntax), however, are not reported. The authors may consider adding the search strings either to the main text or the Supplementary material.

Response: We agree with the reviewer very much. We put the search strategy into the supplementary materials

[4] Comment: In the Results section, I found the summary of findings regarding the participants tested in the different studies a bit short. I suggest a more detailed elaboration of the samples tested in the previous studies. Were there under-, or post-graduate students tested? Do the studies reviewed report any information about the students’ education type? I feel this question relevant, because, for example, medical students may have been even more affected by the pandemic than students enrolling other type of courses.

Response: We strongly agree with the reviewer's suggestion, but unfortunately, after carefully browsing these included studies, we did not find any minors included, and the article did not explain the education type of college students. We will definitely pay attention to this information in our future research.

[5] Comment: According to the inclusion criteria only those studies were selected where the students tested were afflicted by negative emotions during the COVID-19 pandemic. Is there any information that the negative emotions encountered were specifically related to COVID-19? Were, for example, any COVID-specific anxiety or depression assessments made in the studies reviewed, or the emotions were assessed by general methods (e.g. general depression, anxiety inventories) only?

Response: We would like to thank the reviewer for his comments. He has done many studies on the impact of COVID-19 and negative emotions, such as "Psychiatric disorders in health professionals during the COVID-19 pandemic: A systematic review with meta-analysis, in which only general methods (e.g., general depression and anxiety lists) are used to assess mood.

[6] Comment: The COVID-19 pandemic is typically characterized with consecutive waves. If this information is available in the 8 studies reviewed, it may also be informative to indicate under which COVID wave(s) the psychological interventions were performed.

Response: We strongly agree with the reviewer's opinion, but we can't extract the information about the novel coronavirus from the article. However, the reviewer has given us some inspirations, which we will definitely pay attention to in future research.

[7] Comment: The authors used an appropriate method – the Cochrane Handbook for Systematic Reviews of Interventions – to assess the potential risk of biases in the studies reviewed. Although the results of the quality assessment are clearly presented in Figure 2-3, the authors should also summarize their findings for quality and bias assessment in text (e.g. in the same section where the findings for publication bias are reported).

Response: We are very grateful for the reviewer's valuable comments. We have described the quality evaluation of the article in general.

[8] Comment: Finally, regarding the Discussion section, I have one question only. The authors explain the finding of the non-significant result for the somatic symptoms with that of “this might be associated with the number of studies included”. I found this statement unclear. Why would the number of studies included affect the intervention-outcome for somatic symptoms, or more specifically, why would it affect somatic symptoms more than other symptoms (depression and anxiety)?

Response: The small number of studies included and the small sample size involved will become important factors affecting the credibility of our conclusions, therefore, we can also say so.

 

Reviewer #2:Overall:

1. Comment: Looking into the emotional effects of Covid-19. This is something not often spoken about in society and something which holds merit and importance. Particularly the idea of abandonment, desperation, incapability and exhaustion.

Looking into the university age demographic and how they are more susceptible to the emotional effects of Covid-19 due to the different conditions when at school and away from school.。

Response: Thank you very much for the reviewer's opinion, college students a lot because of the outbreak risk control at home now, or can't contact to the classroom, network classroom efficiency is very low, so the students will suffer many academic pressure, and because the contemporary is the Internet age, there are a lot of negative information spread very quickly, as a result of psychological health of college students suffer from huge challenge. It doesn't make a difference just because you leave school at a different time.

2. Comment: The theory proposed here is interesting and has the potential to show clear signs that psychotherapy has a positive impact on helping deal with the emotional strain of the Covid-19 pandemic. However, I think that parts of this theory are generic and rather broad. The theory suggests that psychotherapy could help with symptoms such as depression and anxiety brought on by the pandemic. Whilst this does hold some merit, I feel like this is true of depression and anxiety which is caused by any number of reasons besides the pandemic. It has been widely accepted that psychotherapy has a positive impact on anxiety and depression as a whole and I would have liked a little more data on the difference between the anxiety and depression felt as a result of the pandemic against the anxiety and depression felt in everyday life. Is the end result of anxious and depressed feelings still the same as before the pandemic or are we talking about a new and differing feeling as a result of Covid-19?

Response: We thank you very much for the opinions of the referees, there have been studies that show "10.1177/00207640211003121." before the new crown outbreak of a pandemic will increase negative emotions (anxiety, depression, and psychological distress), so early intervention is required. So the negative emotions we're talking about with the COVID-19 pandemic are different from normal negative emotions.

3. Comment: The link between anxiety and depression with the increase of infection. This was mentioned but not in depth and I think a little more detail regarding this would have helped give the paper more meaning and depth.

Response: Thanks for the comments of reviewers, we added the link between anxiety, depression and increased infection in the discussion section.

Meta-analysis/ results/method:

4. Comment: The analysis itself was well undertaken. I'd recommend that the offers incorporate the following into the paper to strengthen the work itself but otherwise I find it well written:

Overall, I would have suggested that the authors use another Effect Size measurement (such as Cohen's D) or Hedge's G for the measurement. Since it is not used here, i would employ authors to explain why it is not used.

Response: We are very grateful to the reviewers for their comments. We have added them to our article,

5. Comment: -I'd recommend authors employee the Begg and Egger tests to further elaborate on the study findings. The Begg and Egger test usually is done after an asymmetrical funnel plot is undertaken. Also, if authors use such you can find how many studies are needed to negate the present effect.

Response: We conducted the egger test for each anxiety, depression, stress and other indicators, and found that anxiety p=0.046, depression p=0.801, stress p=0.355, P>0.05, indicating that there was no publication bias, and P<0.05, indicating that there was publication bias.

6. Comment: -More within study differences would have been nice to see and I recommend authors add those in to the analysis. For example, authors could have seen the effect of gender, sex, medication status, education level, differences between countries, comorbidites. These could have strengthened the paper.

Response: We strongly agree with the reviewer's suggestion, but unfortunately, after carefully browsing these included studies, we did not find any minors included, and the article did not explain the education type of college students. We will definitely pay attention to this information in our future research.

7. Comment: -I'd also recommend that the authors add in Scopus and PsychInfo as a way for them to add in more studies. As well as cross-checking between articles for more studies

Response: Thank the reviewers for their comments. We added the corresponding database and re searched it. But no new articles can be included,

 

#Editor

Gabriella Vizin, PhD

Journal Requirements:

1.Comment: Please ensure that your manuscript meets PLOS ONE's style requirements, including those for file naming. The PLOS ONE style templates can be found at 

Response: Thank you very much for your comments. We have adjusted the format of the article as required.

2.Comment: Thank you for stating the following financial disclosure: 

"This work was supported by the National Natural Science Foundation of China (Grant No.82102600)."

Response: Thank you very much for your comments. We have made adjustments as required. 

3.Comment: Thank you for stating the following in the Funding Section of your manuscript: 

"This work was supported by the National Natural Science Foundation of China (Grant No.82102600)"

"This work was supported by the National Natural Science Foundation of China (Grant No.82102600)."

Response: Thank you very much for your comments. We have made adjustments as required. 

4.Comment: PLOS requires an ORCID iD for the corresponding author in Editorial Manager on papers submitted after December 6th, 2016. Please ensure that you have an ORCID iD and that it is validated in Editorial Manager. To do this, go to ‘Update my Information’ (in the upper left-hand corner of the main menu), and click on the Fetch/Validate link next to the ORCID field. This will take you to the ORCID site and allow you to create a new iD or authenticate a pre-existing iD in Editorial Manager. Please see the following video for instructions on linking an ORCID iD to your Editorial Manager account: https://www.youtube.com/watch?v=_xcclfuvtxQ.

Response: Thank you very much for your comments. We have handled them as required.

---

## [Editor Report · Decision Letter 1]

15 Nov 2022

PONE-D-22-25934R1Psychological Intervention for Negative Emotions Aroused by COVID-19 Pandemic in University Students: A Systematic Review and Meta-AnalysisPLOS ONE

Dear Dr. Shao,

Thank you for submitting your manuscript to PLOS ONE. After careful consideration, we feel that it has merit but does not fully meet PLOS ONE’s publication criteria as it currently stands. Therefore, we invite you to submit a revised version of the manuscript that addresses the points raised during the review process.

We look forward to receiving your revised manuscript.

Kind regards,

Gabriella Vizin, PhD

Academic Editor

PLOS ONE

Journal Requirements:

Additional Editor Comments:

Overall, this is a very interesting, important study with relevant findings on the importance of the CBT-based interventions for university students during the COVID-19 period.

Thank you for revising the manuscript based on the reviewers' comments.

However, I have a few other suggestions for improving your manuscript for publication.

Abstract: please detail the types of psychotherapies (for example: CBT-based interventions).

Introduction: please, don't use the word "hot-spot” (line 48), try to find another sophisticated word.

line 52-53: This sentence is not necessary: "Psycho-behavioral therapies mainly refer to motive intervention, cognitive-behavioral therapy, behavioral intervention, aversion therapy, and abstain addiction therapy", but you should define the types of the mentioned psychotherapies (all CBT-based interventions).

And you should define not only mindfulness, but also CBT and DBT in this part of introduction. Anyway, these are evidence-based interventions, so you can mention this fact in this section.

Please correct the mentioned problems in your manuscript.

---

## [Author Response · Author response to Decision Letter 1]

17 Nov 2022

List of Responses

Dear Editors

Thank you for your letter and comments concerning our manuscript entitled “Psychological Intervention for Negative Emotions Aroused by COVID-19 Pandemic in University Students: A Systematic Review and Meta-Analysis” (ID: PONE-D-22-25934R1). Those comments are all valuable and very helpful for revising and improving our paper, as well as the important guiding significance to our researches. We have studied comments carefully and have made correction which we hope meet with approval. Revised portion are marked in red in the paper. The main corrections in the paper and the responds to the reviewer’s comments are as flowing:

Responds to the comments:

Additional Editor Comments:

Overall, this is a very interesting, important study with relevant findings on the importance of the CBT-based interventions for university students during the COVID-19 period.

Thank you for revising the manuscript based on the reviewers' comments.

1.Comment: However, I have a few other suggestions for improving your manuscript for publication.

Abstract: please detail the types of psychotherapies (for example: CBT-based interventions).

Responds: We are very much in agreement with the editor's request, we have made changes to the article.

2.Comment: Introduction: please, don't use the word "hot-spot” (line 48), try to find another sophisticated word.

Responds: I'm really sorry that we didn't make it clear. We have modified it.

3.Comment: line 52-53: This sentence is not necessary: "Psycho-behavioral therapies mainly refer to motive intervention, cognitive-behavioral therapy, behavioral intervention, aversion therapy, and abstain addiction therapy", but you should define the types of the mentioned psychotherapies (all CBT-based interventions).

Responds: We have modified this part of the article as required.

4.Comment: And you should define not only mindfulness, but also CBT and DBT in this part of introduction. Anyway, these are evidence-based interventions, so you can mention this fact in this section.

Please correct the mentioned problems in your manuscript.

Responds: Thanks very much for the editor's comments, we have revised and briefly introduced all the interventions covered in this article.

 

#Editor

Gabriella Vizin, PhD

Journal Requirements:

1.Comment: Please ensure that your manuscript meets PLOS ONE's style requirements, including those for file naming. The PLOS ONE style templates can be found at 

Response: Thank you very much for your comments. We have adjusted the format of the article as required.

2.Comment: Thank you for stating the following financial disclosure: 

"This work was supported by the National Natural Science Foundation of China (Grant No.82102600)."

Response: Thank you very much for your comments. We have made adjustments as required. 

3.Comment: Thank you for stating the following in the Funding Section of your manuscript: 

"This work was supported by the National Natural Science Foundation of China (Grant No.82102600)"

"This work was supported by the National Natural Science Foundation of China (Grant No.82102600)."

Response: Thank you very much for your comments. We have made adjustments as required. 

4.Comment: PLOS requires an ORCID iD for the corresponding author in Editorial Manager on papers submitted after December 6th, 2016. Please ensure that you have an ORCID iD and that it is validated in Editorial Manager. To do this, go to ‘Update my Information’ (in the upper left-hand corner of the main menu), and click on the Fetch/Validate link next to the ORCID field. This will take you to the ORCID site and allow you to create a new iD or authenticate a pre-existing iD in Editorial Manager. Please see the following video for instructions on linking an ORCID iD to your Editorial Manager account: https://www.youtube.com/watch?v=_xcclfuvtxQ.

Response: Thank you very much for your comments. We have handled them as required. 

 

Reviewer #1

Review about the manuscript entitled as ‘Psychological intervention for negative emotions aroused by COVID-19 pandemic in university students: a systematic review and meta-analysis

Dear authors, thank you for allowing me to review your work.

Summary and Overall evaluation

The study is addressed to the systematic and metanalytic review of previous studies reporting results about psychological interventions conducted in randomized controlled trials for negative emotions induced by COVID-10 pandemic in university students. The authors considered 259 articles for review based on extensive database search, and selected finally 8 studies for the full review. The study used appropriate eligibility criteria for study selection and used adequate metanalytic techniques to analyse the effectiveness of the interventions. The analyses performed suggested a positive influence of psychological interventions on emotional complaints arisen during the COVID crisis. More specifically, the authors concluded that the studies reviewed provided clear evidence for significant effectiveness of psychotherapies in alleviating the anxiety, depression, and stress, but not the somatization symptoms. The study was overall well written; I do not have sever concerns. Please see my detailed comments and questions below. 

[1] Comment: In recent years, many systematic reviews have been published on the effectiveness of psychological interventions for psychological problems caused by the COVID epidemic. How different and specific is the present review compared to the previous ones with similar scope? I suggest that the authors extend the introduction with a comparison of their study with similar reviews published earlier.

Response: We are very grateful for the reviewer's comments. As the reviewer said, there have been many systematic reviews on the effectiveness of psychological intervention on psychological problems caused by COVID-19 in recent years, but previous studies did not pay attention to the special group of college students, and no similar meta-reports have been published.

[2] Comment: Based on the supplementary information (i.e. PRISMA checklist) and the flow diagram of Figure 1, the authors followed the PRISMA guidelines to select the relevant studies. This, however, should also be stated explicitly in the Method section by adding reference(s) relating to PRISMA; e.g. the following one:

Grimshaw et al (2021). The PRISMA 2020 statement: An updated guideline for reporting systematic reviews. Systematic Reviews, 10(1), 89. https://doi.org/10.1186/s13643-021-01626-4

Response: We strongly agree with the reviewer's opinion. We have rewritten prisma and attached it to the attachment.

[3] Comment: The authors report that many databases were searched for the relevant studies. The general or database-specific search strings (syntax), however, are not reported. The authors may consider adding the search strings either to the main text or the Supplementary material.

Response: We agree with the reviewer very much. We put the search strategy into the supplementary materials

[4] Comment: In the Results section, I found the summary of findings regarding the participants tested in the different studies a bit short. I suggest a more detailed elaboration of the samples tested in the previous studies. Were there under-, or post-graduate students tested? Do the studies reviewed report any information about the students’ education type? I feel this question relevant, because, for example, medical students may have been even more affected by the pandemic than students enrolling other type of courses.

Response: We strongly agree with the reviewer's suggestion, but unfortunately, after carefully browsing these included studies, we did not find any minors included, and the article did not explain the education type of college students. We will definitely pay attention to this information in our future research.

[5] Comment: According to the inclusion criteria only those studies were selected where the students tested were afflicted by negative emotions during the COVID-19 pandemic. Is there any information that the negative emotions encountered were specifically related to COVID-19? Were, for example, any COVID-specific anxiety or depression assessments made in the studies reviewed, or the emotions were assessed by general methods (e.g. general depression, anxiety inventories) only?

Response: We would like to thank the reviewer for his comments. He has done many studies on the impact of COVID-19 and negative emotions, such as "Psychiatric disorders in health professionals during the COVID-19 pandemic: A systematic review with meta-analysis, in which only general methods (e.g., general depression and anxiety lists) are used to assess mood.

[6] Comment: The COVID-19 pandemic is typically characterized with consecutive waves. If this information is available in the 8 studies reviewed, it may also be informative to indicate under which COVID wave(s) the psychological interventions were performed.

Response: We strongly agree with the reviewer's opinion, but we can't extract the information about the novel coronavirus from the article. However, the reviewer has given us some inspirations, which we will definitely pay attention to in future research.

[7] Comment: The authors used an appropriate method – the Cochrane Handbook for Systematic Reviews of Interventions – to assess the potential risk of biases in the studies reviewed. Although the results of the quality assessment are clearly presented in Figure 2-3, the authors should also summarize their findings for quality and bias assessment in text (e.g. in the same section where the findings for publication bias are reported).

Response: We are very grateful for the reviewer's valuable comments. We have described the quality evaluation of the article in general.

[8] Comment: Finally, regarding the Discussion section, I have one question only. The authors explain the finding of the non-significant result for the somatic symptoms with that of “this might be associated with the number of studies included”. I found this statement unclear. Why would the number of studies included affect the intervention-outcome for somatic symptoms, or more specifically, why would it affect somatic symptoms more than other symptoms (depression and anxiety)?

Response: The small number of studies included and the small sample size involved will become important factors affecting the credibility of our conclusions, therefore, we can also say so.

 

Reviewer #2:Overall:

1. Comment: Looking into the emotional effects of Covid-19. This is something not often spoken about in society and something which holds merit and importance. Particularly the idea of abandonment, desperation, incapability and exhaustion.

Looking into the university age demographic and how they are more susceptible to the emotional effects of Covid-19 due to the different conditions when at school and away from school.。

Response: Thank you very much for the reviewer's opinion, college students a lot because of the outbreak risk control at home now, or can't contact to the classroom, network classroom efficiency is very low, so the students will suffer many academic pressure, and because the contemporary is the Internet age, there are a lot of negative information spread very quickly, as a result of psychological health of college students suffer from huge challenge. It doesn't make a difference just because you leave school at a different time.

2. Comment: The theory proposed here is interesting and has the potential to show clear signs that psychotherapy has a positive impact on helping deal with the emotional strain of the Covid-19 pandemic. However, I think that parts of this theory are generic and rather broad. The theory suggests that psychotherapy could help with symptoms such as depression and anxiety brought on by the pandemic. Whilst this does hold some merit, I feel like this is true of depression and anxiety which is caused by any number of reasons besides the pandemic. It has been widely accepted that psychotherapy has a positive impact on anxiety and depression as a whole and I would have liked a little more data on the difference between the anxiety and depression felt as a result of the pandemic against the anxiety and depression felt in everyday life. Is the end result of anxious and depressed feelings still the same as before the pandemic or are we talking about a new and differing feeling as a result of Covid-19?

Response: We thank you very much for the opinions of the referees, there have been studies that show "10.1177/00207640211003121." before the new crown outbreak of a pandemic will increase negative emotions (anxiety, depression, and psychological distress), so early intervention is required. So the negative emotions we're talking about with the COVID-19 pandemic are different from normal negative emotions.

3. Comment: The link between anxiety and depression with the increase of infection. This was mentioned but not in depth and I think a little more detail regarding this would have helped give the paper more meaning and depth.

Response: Thanks for the comments of reviewers, we added the link between anxiety, depression and increased infection in the discussion section.

Meta-analysis/ results/method:

4. Comment: The analysis itself was well undertaken. I'd recommend that the offers incorporate the following into the paper to strengthen the work itself but otherwise I find it well written:

Overall, I would have suggested that the authors use another Effect Size measurement (such as Cohen's D) or Hedge's G for the measurement. Since it is not used here, i would employ authors to explain why it is not used.

Response: We are very grateful to the reviewers for their comments. We have added them to our article,

5. Comment: -I'd recommend authors employee the Begg and Egger tests to further elaborate on the study findings. The Begg and Egger test usually is done after an asymmetrical funnel plot is undertaken. Also, if authors use such you can find how many studies are needed to negate the present effect.

Response: We conducted the egger test for each anxiety, depression, stress and other indicators, and found that anxiety p=0.046, depression p=0.801, stress p=0.355, P>0.05, indicating that there was no publication bias, and P<0.05, indicating that there was publication bias.

6. Comment: -More within study differences would have been nice to see and I recommend authors add those in to the analysis. For example, authors could have seen the effect of gender, sex, medication status, education level, differences between countries, comorbidites. These could have strengthened the paper.

Response: We strongly agree with the reviewer's suggestion, but unfortunately, after carefully browsing these included studies, we did not find any minors included, and the article did not explain the education type of college students. We will definitely pay attention to this information in our future research.

7. Comment: -I'd also recommend that the authors add in Scopus and PsychInfo as a way for them to add in more studies. As well as cross-checking between articles for more studies

Response: Thank the reviewers for their comments. We added the corresponding database and re searched it. But no new articles can be included,

---

## [Decision Letter · Decision Letter 2]

6 Mar 2023

Psychological Intervention for Negative Emotions Aroused by COVID-19 Pandemic in University Students: A Systematic Review and Meta-Analysis

PONE-D-22-25934R2

Dear Dr. Shao,

We’re pleased to inform you that your manuscript has been judged scientifically suitable for publication and will be formally accepted for publication once it meets all outstanding technical requirements.

Kind regards,

Kyoung-Sae Na, M.D., Ph.D.

Academic Editor

PLOS ONE

Additional Editor Comments (optional):

Reviewers' comments:

Reviewer's Responses to Questions

**Comments to the Author**

1. If the authors have adequately addressed your comments raised in a previous round of review and you feel that this manuscript is now acceptable for publication, you may indicate that here to bypass the “Comments to the Author” section, enter your conflict of interest statement in the “Confidential to Editor” section, and submit your "Accept" recommendation.

Reviewer #1: All comments have been addressed

2. Is the manuscript technically sound, and do the data support the conclusions?

Reviewer #1: Yes

3. Has the statistical analysis been performed appropriately and rigorously? 

Reviewer #1: Yes

4. Have the authors made all data underlying the findings in their manuscript fully available?

Reviewer #1: Yes

5. Is the manuscript presented in an intelligible fashion and written in standard English?

Reviewer #1: Yes

6. Review Comments to the Author

Reviewer #1: (No Response)

7. PLOS authors have the option to publish the peer review history of their article (what does this mean?). If published, this will include your full peer review and any attached files.

Reviewer #1: No

---

## [Editor Report · Acceptance letter]

8 Mar 2023

PONE-D-22-25934R2 

Psychological Intervention for Negative Emotions Aroused by COVID-19 Pandemic in University Students: A Systematic Review and Meta-Analysis 

Dear Dr. Shao:

I'm pleased to inform you that your manuscript has been deemed suitable for publication in PLOS ONE. Congratulations! Your manuscript is now with our production department. 

Kind regards, 

on behalf of

Dr. Kyoung-Sae Na 

Academic Editor

PLOS ONE